# A Study of Visual Perception Based on Colour and Texture of Reconstituted Decorative Veneer



**Ting Huang** [1,2], **Chengmin Zhou** [1,2,*], **Xiaomeng Wang** [1,2] **and Jake Kaner** [3]

1 College of Furnishings and Industrial Design, Nanjing Forestry University, Nanjing 210037, China; tinghuang@njfu.edu.cn (T.H.); xmw_ch@163.com (X.W.)

2 Jiangsu Co-Innovation Center of Efficient Processing and Utilization of Forest Resources, Nanjing 210037, China

3 School of Art and Design, Nottingham Trent University, Nottingham NG1 4FQ, UK; jake.kaner@ntu.ac.uk

* Correspondence: zcm78@163.com

**Abstract:** Color and texture, as vital physical attributes of recombinant decorative thin wood, represent the initial visual information perceived by the human eye. These elements play a crucial role in shaping the human viewing experience. This paper centers on the user's visual perception of recombinant decorative thin wood and is divided into two main sections: the extraction and analysis of color and texture features, and experimental research evaluating combinations of color and texture. The experiments yielded conclusions indicating that the design of color and texture significantly influences objective eye movement data. Specific findings include: (1) The gaze duration, gaze counts, and hotspot maps of the subjects consistently demonstrate high agreement across the three eye movement indicators. Notably, a significant difference is observed between gaze indicators and color blending modes. (2) Asymmetric oblique blending and high-brightness circumferential blending tend to attract subjects' eye attention more effectively. These color groups are characterized by transparent layers, increased brightness, and a pronounced visual impact. Such features enhance the design effect of the texture, highlighting its layers.

**Keywords:** reconstituted decorative veneer; visual perception; color clustering; texture analysis; color blending

## 1. Introduction

In recent years, the consumer market for home furnishings has displayed robust potential, propelling the rapid growth of the furniture industry and creating a substantial demand for wood and wood products [1]. In a bid to safeguard forest resources, preserve the ecological environment, and foster a harmonious coexistence of humanity and nature, China initiated an increased emphasis on green homes in 2012. The country introduced a series of policies aimed at promoting the low-carbon home furnishing industry, aligned to top-level planning perspectives. Artificial fast-growing timber, due to its softwood and low density, cannot be directly employed as raw wood products [2]. It requires modification to enhance its performance and increase stability. This quality-improved artificial timber exhibits robust functionality, has broad applications, and commands high added value [3]. Recognized as a green and renewable material, Reconstituted Decorative Veneer (RDV), also known as reconstituted wood or the simulation of precious wood, utilizes the principles of bionics. Ordinary wood undergoes advanced processing, restructuring, and landscaping treatments to produce new all-wood materials with superior performance. This approach has garnered international consensus, particularly against the backdrop of energy conservation, emission reduction, and global responses to climate change [4]. RDV, which has been subjected to artificial wood modification treatments, offers the wood substrate enhanced visual possibilities. This aids in optimizing its performance, effectively meeting the market's demand for ample timber resources and elevating environmental

standards in home materials. As the home furnishing industry undergoes transformation and upgrades, RDV is ready to encounter numerous market opportunities [5–7].

Simultaneously, with the younger generation emerging as the predominant consumer demographic, the home furnishing sector has progressively adopted a design trend centered around sensory experiences. This trend caters to the comfort requirements of both the older and newer iterations of consumers. The newer consumer groups, in particular, seek a personalized home environment while also prioritizing overall style and the value of furnishings. Sensory comfort has become a pivotal factor in their decision-making process. The native texture and color of wood materials play a crucial role in generating a sense of empathy and providing a comfortable feeling, both visually and psychologically. In recent years, scarcity in timber resources and escalating prices of wood furniture have led some consumers to compromise on purchasing solid wood products. Instead, they opt for furniture items that emulate solid wood decorative effects. These products rely on techniques such as natural wood grain scanning collection and processing for surface decoration. This includes methods like direct printing of wood grain, wood grain-impregnated paper veneer, and digital wood grain 3D printing. However, a segment of the current market features imitation veneer, wood grain-impregnated paper veneer, digital printing wood grain, and other wood grain techniques that suffer from issues such as unclear patterns, inappropriate color contrasts, and inconsistent visual effects. Consequently, the resulting products lack authentic wood texture and exhibit monotonous colors. This compromises their ability to meet the psychological needs of individuals seeking sensory comfort in indoor spaces.

Existing research on the visual perception of wood materials can be broadly categorized into three focal points, as identified by some scholars: (1) the visual sensations elicited by wood materials; (2) attitudes and preferences (aesthetic evaluation) towards various wood products and (3) emotional and psychophysiological responses to wood perceptions [8]. Burnard et al. (2017) [9] conducted an assessment of the disparities in people's preferences for wood and non-wood environments. Their findings revealed that wood environments received higher audience and preference evaluations. Delving into the literature, Bringslimark et al. (2010) [10] highlighted that the texture, knots, and timber color of wood materials are crucial surface visual attributes influencing people's assessment of wood material preferences. It is asserted that factors such as wood species, knots, wood coverage, application form, and spatial location relationships can also impact preference assessments. All these studies underscore that the visual characteristics of timber materials, including color, texture, and luster, directly manifest in the external visual performance of the products to which they are applied. These characteristics serve as a critical reference for users in evaluating a product.

Wood color stands out as the pivotal feature that reflects both the visual and psychological aspects of a material's surface. In the realm of wood and wood products, wood color serves as the primary characteristic. It not only plays a crucial role in assessing wood quality but also serves as a fundamental criterion for measuring the value of a product. The significance of wood color extends to the design and production of wood products. Traditionally, researchers employed color names or combinations of color names to describe the surface color of wood. The quantitative characterization of wood color; however, became feasible with the development of chromaticity research. Since the 1960s, scientists in the United States, Europe, and Japan have conducted thorough investigations into the quantitative measurement of wood color, yielding fruitful results. Scholars worldwide have consistently highlighted that visual physical quantities, such as color, are primary factors influencing users' emotional preferences, cognitive load, and eye movements [11]. As a background material in indoor environments, the visual characteristics of the color of wood materials become a crucial factor affecting people's psychological impressions [12,13]. The color is dependent on the timber color of the substrate and exerts a certain degree of influence on the texture. Jafarian et al. (2018) [14] explored the relationship between the color of wood veneer and space lighting, concluding that wood veneer with bright colors contributes to enhancing the quality of space lighting, thereby creating a visually

comfortable indoor atmosphere. The surface gloss of wood, its light reflection properties, and anisotropy collectively create a varied visual effect. Research has shown that different species of finish gloss, through careful interaction, can produce the desired visual impact for users, thereby enhancing the quality of indoor space and improving lighting ambiance [15]. Minoru (1987) [15] found that wood knots and texture, as natural visual physical quantities in wood decorative material, positively impact the visual evaluation of wood. Subsequent studies by Matsumoto et al. (2016) [16], Broman (2001) [17], and Manuel et al. (2016) [18] have reinforced the strong correlation between preference evaluation and the number of knots. Haemaelaeinen et al. (2012) [19] utilized eye-tracking techniques to experimentally assess the contrast difference on the surface of wood material due to painting, demonstrating a correlation between texture clarity under the influence of coating sheen and the attractiveness and contrast values of the wood material's surface.

While some studies have affirmed that analyzing a single surface visual, physical quantity of wood can measure its visual properties, the diverse applications of wood on spatial surfaces necessitate a nuanced understanding of how multivariate interactions influence people. Moreover, variations in geographical, national, socio-economic, and cultural contexts can lead to differing perceptions of wood materials. Joel et al. (2019) [20] utilized a semantic differential approach to assess perceptual preferences for visual and haptic aspects of wood materials among Swedish and Japanese users. They found that Japanese users preferred homogenous wood surfaces, while Swedish users favored wood with textured or knotted surfaces. Ramananantoandro et al. (2013) [21] discovered that users in Antananarivo, Madagascar, exhibited a significant preference for yellow, darker wood timber colors, based on the CIELab* color model with sensory analysis. They highlighted socio-economic factors as influencing color preferences. Strobel et al. (2017) [22] asserted that cultural expectations can also significantly impact the perception of wood materials. When evaluating people's physiological responses to the visual properties of wood-based materials, commonly used indicators include brain activity, autonomic activity, endocrine activity, and immune system activity. Early overseas research often used blood pressure, heart rate, and pulse rate as physiological evaluation indices. Takeshi et al. (1998) [23] conducted experiments demonstrating that wood affects pulse rate and systolic blood pressure changes more than silk and other materials. Subsequent studies (Satoshi, 2008 [24]; Yuko et al., 2002 [25]) combined human heart rate, blood pressure, and subjective evaluation indices, concluding that individuals in spaces with wooden decorative backdrops had significantly lower blood pressure and fewer negative emotions than those facing backdrops made of other materials. In addition to physiological signals, the collection of eye movements and EEG, skin conductivity, salivary cortisol, heart rate variability, and other physiological indicators are gradually becoming evaluation indices for determining visual perception of wood materials [26]. Infrared-based non-contact perception determination methods are also under further development. Visual perception, distinct from vision, encompasses not only the direct stimuli produced by visual senses to the external world but also invokes memories and experiences stored in the brain [27]. The visual properties of wood materials can directly impact human mental health. For instance, changes in wood color can alter perceptions of light and dark, warmth and cold, while changes in texture can influence perceptions of strength and weakness, likes and dislikes [28,29]. Wang et al. (2015) [30] utilized the semantic differential method to divide college students' sensory properties and quantitative evaluations of 28 finish patterns into three dimensions. The results indicated that patterns with rich color variations differed in visual properties, providing a more sensual and elegant feeling in the sensory dimension. Over 90% of decorative veneers were considered to possess "natural-practical" and "man-made-decorative" characteristics in the evaluation dimension [31–34].

### 1.1. Contributions

- Enriching theoretical and experimental research on visual perception in veneer. There needs to be more studies on visual perception in veneer. In this study, we investigate

qualitatively and quantitatively the relationship between color and visual perception of reconstituted decorative veneer and gain a deeper understanding of the logic of visual perception as an evaluation criterion in the design of home finishes. The design and application strategies and key points are sorted out further to expand the theory of visual perception in experimental research.

- It is improving RDV's decorative properties and visual aesthetics. Visual perception is an essential part of the study of RDV, involving the visual sensory experience and aesthetic performance of its application products and environment. It directly affects the spatial atmosphere people perceive and is closely related to the attractiveness of home products. Exploring the visual perception characteristics of RDV plays a crucial role in improving the visual aesthetics and decorative performance of veneers.

- Promote the application of reconstituted decorative lamellas and other new materials and technologies in home décor. Materials and their decorative properties are the most essential material support for creating a comfortable environment. Developing new finishing materials can promote the design innovation of home furnishing products and enrich the varieties of home furnishing materials. To transform and upgrade the home furnishing industry, it should shift from the pure pursuit of precious wood to developing new alternative materials. Therefore, it is necessary to systematically study the restructuring of decorative materials, develop home finishing products that are popular in the market, and create conditions for the large-scale application of reconstituted decorative veneer in the home industry. At the same time, it is also of reference significance for using other new materials for home decoration.

### 1.2. Paper Outline

The purpose of this paper is to explore the visual perceptual relationship between color and texture in RDV. Specific sections are described below: The second chapter focuses on the methodology, which is based on the theory of color blending and sample clustering for the color and texture of RDV. Section 3 focuses on a set of visual perception experiments with RDV: selecting 22 texture types and varying their color palettes for eye-tracking experiments. Section 4 analyses and discusses the experimental data. Section 5 concludes the study.

## 2. Methodology

### 2.1. Quantitative Analysis of the Color of RDV

The choice of color attributes, application location, area ratio and adjacency will impact the overall visual effect of the product. In order to obtain the color composition of RDV, the color extraction of RDV was carried out to quantify the color information and provide a completely independent and objective evaluation of the RDV image.

The fundamental methodology employed in this study is delineated as follows: Firstly, the color extraction technique is applied to discern the primary colors present in the collected samples. Subsequently, the outcomes of the color extraction are visualized through a color network, where nodes represent the primary colors identified. This is followed by a detailed analysis of the color modulation mode and the characteristics of color attributes. In the final step, the color distribution characteristics of existing RDV products are consolidated and summarized. To facilitate a comprehensive analysis of the color distribution in RDV, the research incorporates theories and methods for color analysis, extraction, and expression. The selection of these theories and methods is informed by a comparative analysis of color science and related methodologies, ensuring a robust foundation for the study's approach to understanding the color features of RDV.

The RDV images utilized in this study were predominantly sourced from official brand websites and various media data platforms. Notably, the selected images were captured from flat and downward angles. To create the experimental sample library of RDV, image pre-processing was conducted using Adobe Photoshop 2021 software. The processed images were standardized to a uniform size of 1440 pixels × 1132 pixels, with a

resolution exceeding 300 dpi. In order to validate the color accuracy and bridge the gap between electronic images and physical samples, a meticulous process was undertaken. Twelve RDV images were randomly sampled from the experimental library, and these were then compared with corresponding physical samples procured in advance from various brands. The physical samples were identical to the images used for color information comparison. The objective of this step was to ensure the fidelity of color representation in subsequent experiments. The specific steps involved in the color information comparison are outlined as follows: (1) Using a color picker to select the color of each of the 12 RDV objects, extract and record the Lab value of each color. (2) Extract the RGB values of the colors of the corresponding RDV images by the algorithm. (3) The color values extracted by the algorithm are converted into $L^*a^*b^*$ values. The color difference $\Delta E$ between the color values of the color picker and the color values obtained by the algorithm is calculated using the CIELAB color difference formula, i.e., Equation (1), and the specific data are shown in Table 1.

$$\Delta E = \sqrt{(l_1^* - l_2^*)^2 + (a_1^* - a_2^*)^2 + (b_1^* - b_2^*)^2} \tag{1}$$

$l_1^*, l_2^*$—The lightness. The darkest black at $L^* = 0$, the brightest white at $L^* = 100$.

$a_1^*, a_2^*$—The red/green opponent colors. Green at negative $a^*$ values, red at postive $a^*$ values.

$b_1^*, b_2^*$—The yellow/blue opponent colors. Blue at negative $b^*$ values, yellow at postive $b^*$ values.

**Table 1.** Data of color values of physical objects and color values extracted by algorithms.

| Style | Number | Color 1 | | | Color 2 | | | Color 3 | | |
|---|---|---|---|---|---|---|---|---|---|---|
| | | *L\** | *a\** | *b\** | *L\** | *a\** | *b\** | *L\** | *a\** | *b\** |
| Color picker | ALPI-A12 | 52 | −1 | 9.4 | 29.6 | 0.8 | 6.4 | 74.8 | −3.2 | 14.2 |
| | ALPI-A9 | 41.7 | 30.2 | 35.8 | 35.8 | 27 | 28.3 | 19.6 | 2.4 | 7.3 |
| | LATHO-W7 | 24 | −0.2 | 3.4 | 36 | 3.5 | 13.6 | 37.5 | 0.2 | 3.3 |
| | ALPI-A11 | 74.4 | −0.2 | 15.3 | 53.3 | 0.7 | 12.4 | 37.1 | 3.5 | 9.7 |
| | SP-W15 | 27 | −1.1 | 7.8 | 49 | 0.6 | 18.6 | 43.5 | 5.3 | 16.4 |
| | TABU-W154 | 60.1 | 11.3 | 41 | 53.8 | 14.4 | 39.4 | 62.6 | 15 | 39.1 |
| | LATHO-W10 | 23.9 | 9.4 | −1.3 | 34.6 | 3.4 | 8.5 | 39.5 | 1 | 8.1 |
| | TUBAO-W57 | 25.5 | 0.5 | 4.3 | | | | | | |
| | ALPI-W12 | 31.1 | −11.9 | −7.8 | | | | | | |
| | TUBAO-W99 | 58.7 | 6.7 | 23.8 | | | | | | |
| | ALPI-W3 | 25.4 | 15.7 | 5.2 | | | | | | |
| | ALPI-A39 | 19.6 | 7.3 | −15.4 | | | | | | |
| Arithmetic | ALPI-A12 | 48.3 | 2.5 | 2.8 | 28.4 | 1.9 | 0 | 70.9 | 1.6 | 5.8 |
| | ALPI-A9 | 38.9 | 32.8 | 30.2 | 34.7 | 25.3 | 25 | 18.3 | 1.2 | 2.6 |
| | LATHO-W7 | 21.8 | 2.7 | −1.9 | 34 | 6.7 | 5.8 | 35.4 | 2.9 | −2.9 |
| | ALPI-A11 | 71.8 | 2.4 | 8.3 | 51.1 | 2.7 | 4.7 | 35.5 | 2.7 | 2.9 |
| | SP-W15 | 25.1 | 1.1 | −0.3 | 46.1 | 4.8 | 10.5 | 40.9 | 4.9 | 10.8 |
| | TABU-W154 | 56.7 | 14.8 | 36.7 | 52.4 | 13.8 | 35.5 | 61.4 | 15 | 34.8 |
| | LATHO-W10 | 23.7 | 4.7 | 2.4 | 32.7 | 5.3 | 2.6 | 37.1 | 3.9 | 2.1 |
| | TUBAO-W57 | 23.6 | 0 | 1.4 | | | | | | |
| | ALPI-W12 | 28.3 | −11 | −7.6 | | | | | | |
| | TUBAO-W99 | 61.7 | 6 | 23.9 | | | | | | |
| | ALPI-W3 | 23.5 | 15.4 | 5.1 | | | | | | |
| | ALPI-A39 | 18.7 | 8.4 | −20.3 | | | | | | |

Table 2 shows the difference between the color values of the RDV object extracted by the color picker and the image extracted by the algorithm, most of which are around five and located below 10. The $\Delta E$ between the two colors is less than 6.5, which indicates that the human eye is unable to identify the color difference between the two, thus verifying that the electronic image downloaded from the brand's official website can be used as a substitute for the RDV object for the color extraction study.

**Table 2.** The color value of real objects and color difference data of color value extracted by the algorithm.

| Number | Color 1 | Color 2 | Color 3 |
|---|---|---|---|
| ALPI-A12 | 6.89 | 5.63 | 7.47 |
| ALPI-A9 | 3.81 | 1.52 | 3.67 |
| LATHO-W7 | 5.87 | 6.42 | 6.49 |
| ALPI-A11 | 4.89 | 5.62 | 4.88 |
| SP-W15 | 6.72 | 6.13 | 3.52 |
| TABU-W154 | 3.53 | 1.56 | 2.08 |
| LATHO-W10 | 4.75 | 5.52 | 6.12 |
| TUBAO-W57 | 2.63 | | |
| ALPI-W12 | 1.57 | | |
| TUBAO-W99 | 3.04 | | |

Specific criteria were established to meet data quality standards for excluding abnormal images from the samples. Images affected by evident brand post-processing, substantial alterations in light and brightness, and incorrect shooting angles were excluded. This meticulous selection process was implemented to ensure the accuracy of color information and the extraction of relevant data. After conducting the necessary statistical analysis and applying the defined criteria, a total of 1627 stained imitation natural veneer sample images (as depicted in Figure 1) and 280 reconstituted artistic modeling veneer sample images (as illustrated in Figure 2) were compiled. In aggregate, 1907 veneer sample images were included in the study. These selected images form the basis for the subsequent analysis and investigations in the study.

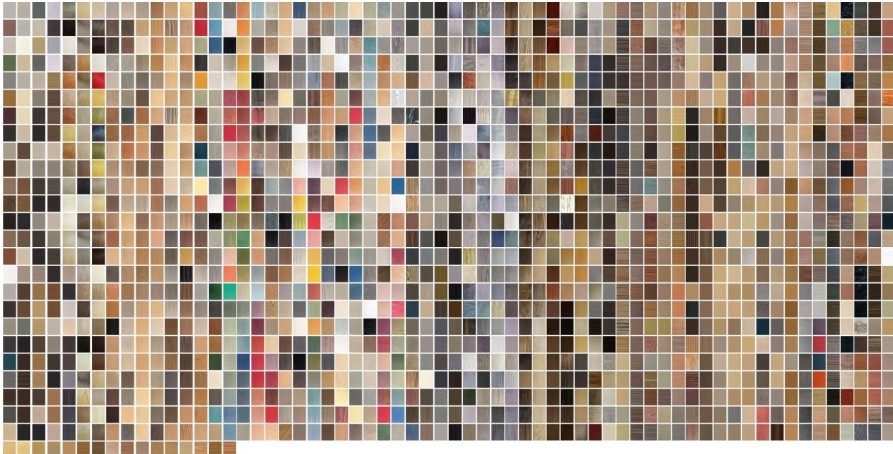

**Figure 1.** 1627 samples of dyed imitation natural veneer.

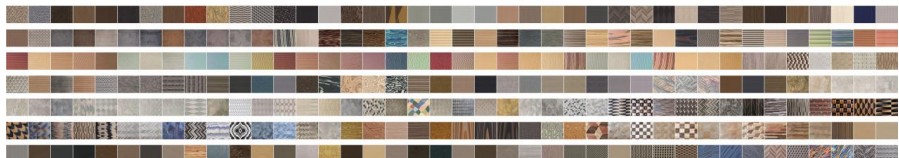

**Figure 2.** 280 samples of engineered artistic veneer.

2.1.1. The Process of color Extraction from Decorative Veneer for Restructuring Artistic Shapes

The color extraction process for reconstituted artistic modeling veneer involves applying the K-means clustering technique to cluster the colors within a single veneer image, specifying the required number of colors. Given the rich color composition of veneer, with transparent primary colors and non-single color elements, the RGB hue mode is chosen

for extracting reconstituted decorative veneer images. Given the variations in reconstitution methods, the color-matching characteristics of veneers can differ. Therefore, the initial clustering center selection is adjusted based on the color-matching effect and pixel composition characteristics of the reconstituted decorative veneer (RDV). This adjustment aims to enhance the accuracy of color extraction by aligning the K value with the image's pixel distribution. The researchers read the colors along the hue ring in hue mode and selected these feature colors as the initial clustering centers. This approach ensures that the color extraction process is attuned to the specific characteristics of the reconstituted artistic modeling veneer, contributing to more accurate and representative results.

As depicted in Figure 3, the K-means algorithm is applied to cluster 276 sample images in a two-step process. The specific experimental steps are outlined as follows: (1) Because the number of color compositions of each veneer is different, a relatively large number of extracted colors are given in the first clustering under the premise of removing the background colors, and several colors will be manually screened as the initial value of clustering in the second clustering. (2) After several rounds of iterative clustering, the primary color composition of each image and the percentage of the situation are obtained. (3) Construct a color network relationship graph based on the extracted color weight and co-occurrence relationships.

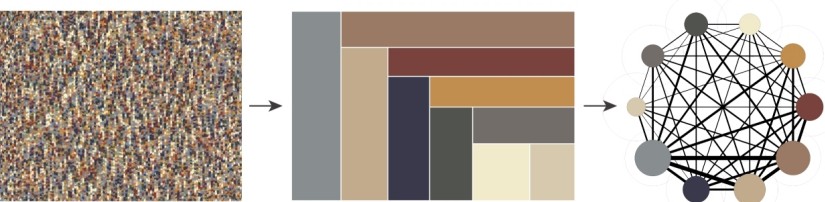

**Figure 3.** Color extraction and color network model construction of single veneer.

2.1.2. Color Extraction Process for Decorative Veneer Imitating Natural Dyeing

Given the substantial number of samples, the color clustering of imitation natural stained veneer is conducted based on the entire sample gallery. The extraction process is divided into two steps: initially, color clustering is performed individually on single reconstituted decorative veneer images, followed by a secondary clustering process based on the comprehensive color map. The gallery color extraction is a secondary extraction based on collecting single-image color extraction results.

As illustrated in Figure 4, the extraction process involves identifying the part of the color that occupies the largest and most representative area in the sample collection of the imitation natural stained veneer image library. This process is a secondary extraction based on all the samples in the library, resulting in the extraction of the dominant and most representative color from the entire collection. The single-image extraction is presented as a result of a rectangle composed of three different proportions of feature colors, forming a new image. In total, 1276 veneer sample images are subjected to color block extraction, and these extracted color blocks are assembled to create a color clustering extraction color library of imitation naturally dyed veneer images. This library comprehensively represents the predominant colors present across the entire spectrum of imitation natural stained veneer samples.

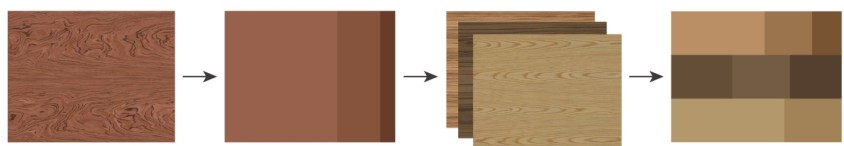

**Figure 4.** The first color clustering of single dyed veneer image.

### 2.1.3. Analysis of Color Clustering Results

(1)  Color Clustering of Reconstituted Artistic Styling Veneer

The color clustering analysis of reconstituted art-shaped veneer involved excluding veneer sample images that might result in the number of clustered colors being less than two due to a single color or excessive blurring. After this exclusion process, a total of 276 veneer sample images with discernible color relationships were obtained through statistical analysis. The distribution of the number of colors in these 276 reconstituted art-shaped veneer samples, along with their respective ratios, is summarized in Table 3.

Among the analyzed samples, it was observed that the maximum number of color combinations in the reconstituted art-shaped veneer did not exceed 10. Specifically: The most prevalent color scheme consisted of three colors, encompassing 112 products, constituting 40.58% of the samples. The second most common was a two-color color scheme, found in 83 products, accounting for 30.07%. Following closely were five-color color schemes, present in 28 products, making up 10.15% of the samples. Four-color color schemes were observed in 21 products, comprising 11.23%. The least common were seven-color color schemes, represented by only 1 product, making up 0.36%.

**Table 3.** Color composition and quantity of 276 art-shaped veneers.

| Number Veneer Colors | Product Number | Percentage |
|---|---|---|
| 2 | 83 | 30.07% |
| 3 | 112 | 40.58% |
| 4 | 21 | 10.15% |
| 5 | 28 | 11.23% |
| 6 | 10 | 3.62% |
| 7 | 1 | 0.36% |
| 8 | 2 | 0.72% |
| 9 | 7 | 2.55% |
| 10 | 2 | 0.72% |

(2)  Color Clustering of Decorative Veneer Imitating Natural Stain

Figure 5 shows the results of the color data extracted based on 1627 images of imitation naturally dyed veneer samples. The color blocks represent the color information extracted from a single image of the veneer, and each block has the same width. In contrast, the different lengths represent different proportions of that color in the image.

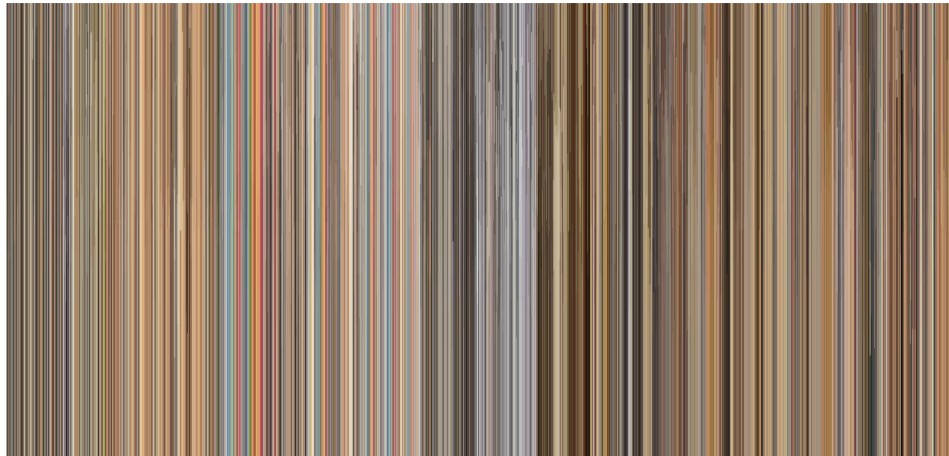

**Figure 5.** The first color clustering of 1627 dyed veneer images.

The number of colors obtained from the primary clustering is relatively large, and secondary clustering is required to derive the most typical colors. K-means clustering is performed on the pooled color library, the mode is set to hue mode, and the proportion of the final extracted colors is considered when extracting the pooled color library colors. The K-means clustering is set to 12 for the imitation natural dyed veneer colors, and these 12 representative colors are enough to assist the design practice in the later stage. Through comparison, it is confident that the extracted colors' output can express the main color characteristics of the existing market imitation natural stained veneer. Specific data are shown in Table 4.

**Table 4.** Clustering data of dyed veneer color.

| Colors | $L^*$ | $a^*$ | $b^*$ | $\Delta H$ | $\Delta L$ | $\Delta C$ | Percentage | Color Block Diagram |
|---|---|---|---|---|---|---|---|---|
| 1 | 68.18 | 4.85 | 21.38 | 034 | 65 | 10 | 21.58% | |
| 2 | 44.72 | 10.30 | 15.23 | 026 | 45 | 11 | 14.86% | |
| 3 | 56.67 | 2.30 | 9.86 | 031 | 54 | 06 | 12.72% | |
| 4 | 33.57 | 5.68 | 15.07 | 034 | 32 | 12 | 11.31% | |
| 5 | 80.01 | 1.18 | 3.61 | 023 | 77 | 02 | 10.55% | |
| 6 | 58.83 | 13.03 | 29.13 | 027 | 57 | 18 | 9.42% | |
| 7 | 22.24 | −1.71 | −0.59 | 091 | 20 | 01 | 4.69% | |
| 8 | 18.81 | 11.50 | 11.23 | 015 | 24 | 10 | 3.92% | |
| 9 | 23.26 | −3.57 | −7.19 | 110 | 23 | 06 | 3.80% | |
| 10 | 97.23 | 0.00 | 0.00 | 037 | 93 | 00 | 2.71% | |
| 11 | 88.24 | 0.31 | 2.32 | 018 | 85 | 01 | 2.42% | |
| 12 | 57.35 | −1.72 | −11.29 | 121 | 54 | 06 | 2.02% | |

Using the Coloro platform to convert the $L^*a^*b^*$ color values to HCL (Hue-Chroma-Luminance) values, which was first proposed by the International Commission on Illumination (CIE). HCL is used more frequently in everyday product color schemes and is especially friendly to the visually impaired and more in line with the human eye's visual system [35,36]. The analysis of its color properties shows that the current domestic and international market of imitation natural dyeing and restructuring of decorative veneer Hue (H) is dominated by warm colors, mainly red and yellow (H: 016~032) and yellow (H: 032~048), a small portion of the hue of the blue (H: 096~112) or blue-violet (H: 112~128). In the terms of Luminance (L) and Chroma (C), medium Luminance and medium Chroma, high Luminance and low color, and low Luminance and medium Chroma predominate.

## 2.2. Quantitative Analysis of the Texture of Reconstituted Decorative Veneer

### 2.2.1. Collection of Texture Types

As the main content of the research in this subsection is the restructuring of the artistic modeling of veneer texture characteristics, in order to avoid the influence of other factors such as color, the images are edited and produced as greyscale images using Adobe Photoshop 2021 software, and some of the sample images are shown in Figure 6.

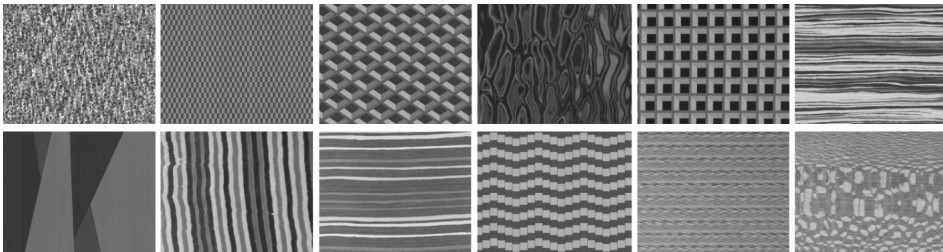

**Figure 6.** Images of some veneer research samples.

### 2.2.2. Extraction of Texture Types

The grain types of RDV are diverse. With the advancement of recombination technology, the design of veneer textures has become more intricate, refined, and diversified. As depicted in Figure 7, the texture types have evolved from simple natural wood grain textures to abstract textures featuring lines, geometry, and other themes. This progression extends to a broad spectrum of natural gradients and 3D effect textures. These variations are combined and matched with specific patterns and abstract processing, resulting in diverse texture effects. As a finishing material, the texture design of reconstituted decorative veneer considers the application area and decorative environment. It typically favors abstract geometric textures or extracts elements from natural textures. This choice ensures that the veneer restructuring assembles textures with aesthetics and continuity in mind. The classification of decorative veneer texture types in the restructuring of the art of modeling veneer can be categorized into three types of texture: (1) The natural wood grain texture embodies the imitation of the wood itself. (2) Imitation of various natural elements, serving as a source for natural elements texture. (3) Various modeling textures are designed for home surface decoration.



**Figure 7.** Texture types of engineered veneer.

Table 5 presents the number and proportion of the three texture types in reconstituted art modeling veneer. Notably, the most substantial proportion is the reconstituted modeling texture, totaling 189 products and accounting for 68.48%; followed by the imitation of natural wood grain texture, totaling 62 and accounting for 22.46% and finally, the imitation of natural elements texture, totaling 25 and accounting for 9.06%. Among these, modeling texture exhibits various combinations of wood, offering rich variations and a strong sense of hierarchy. It possesses the most distinctive design characteristics and decorative value among RDV products. Studying its texture composition and combination characteristics is essential for developing and innovating RDV. Furthermore, it holds a broad application value for subsequent design strategies, specifically in the context of home surface decoration. As a result, this study focuses on the modeling texture belonging to the surface decoration of the home.

**Table 5.** Texture types and quantity proportion of artistic engineered veneer.

| Texture Types | Number | Percentage |
|---|---|---|
| Natural Wood Grain Texture | 62 | 22.46% |
| Faux Natural Elemental Texture | 25 | 9.06% |
| Restructuring Modelling Texture | 189 | 68.48% |
| Total | 276 | |

## 3. Experimental Design

### 3.1. Experimental Methods

The experiments were conducted in a university laboratory with an average ambient lighting intensity of approximately 300 lux. The mainframe operated on Windows 10, and the monitor used was an LCD screen with a resolution of 1920 × 1080 and a refresh rate of 60 Hz. The mean visual distance of the subjects was controlled to be approximately 50 cm. Thirty-eight subjects, aged 20 to 26 years old, with normal or corrected visual acuity and no color blindness, were selected for the experiment. Before the commencement of

the experiment, subjects were required to fill out a basic personal information form and provide open consent for the use of experimental data, acknowledging the experiment's approval by Nanjing Forestry University. Subsequently, they were familiarized with the experimental procedures and given an overview of health and safety precautions.

### 3.2. Stimulus Material

This experimental protocol was designed as a two-factor experiment in which eight colors with different tonal modes were used for visual perception experiments on 22 textures. The two factors were color tonal modes (a) and textures (b). In the color blending mode (a), eight levels were set, as shown in Figure 8. a1~a4 are vertical blending, symmetric oblique blending, asymmetric oblique blending, and radial blending, respectively, and are recorded as color groups a1, a2, a3, and a4; a5 and a6 are shown as the contrast between cool and warm tones under the same spiral blending colors and are recorded as color groups a5, and a6 and a7 and a8 are shown as the contrast between low height and lightness under the circular blending colors, and are recorded as color groups a7 and a8, in turn. Brightness contrast, in turn, is recorded as the color group a7, a8.

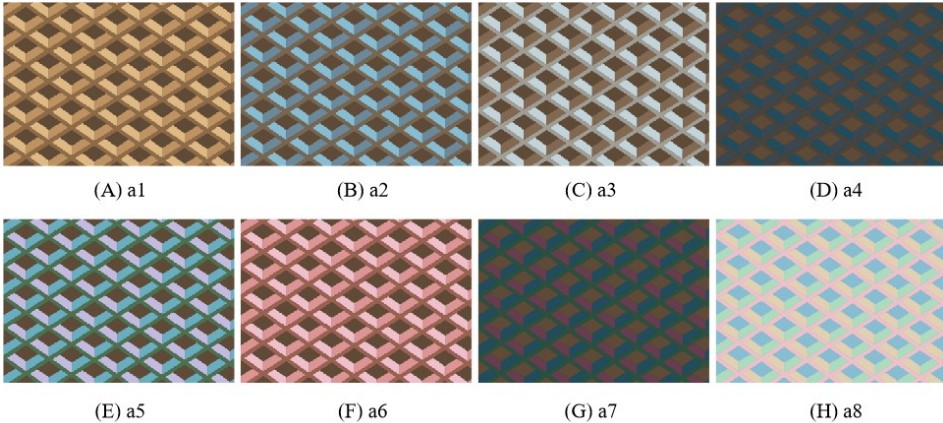

| (A) a1 | (B) a2 | (C) a3 | (D) a4 |
| (E) a5 | (F) a6 | (G) a7 | (H) a8 |

**Figure 8.** Color harmony modes.

In texture (b), 22 levels are set as shown in Figure 9. b1~b22 are sequentially 3D texture-1, 3D texture-2, 3D texture-3, 3D texture-4, Geometric texture-1, Geometric texture-2, Geometric texture-3, Geometric texture-4, Arrow texture-1, Arrow texture-2, Arrow texture-3, Linear texture-1, Linear texture-2, Linear texture-3, Linear texture-4, and Irregular texture-1, Irregular pattern-2, Irregular pattern-3, Irregular pattern-4, Wave pattern-1, Wave pattern-2, Wave pattern-3. The experiment's total number of horizontal combinations is [a1b1, a8b22], 8 × 22 = 176. These 176 level combinations constitute the experimental protocol for the two factors of color mixing mode and texture, so the stimulus material for this experiment was 176 sample images without repeated level combinations. The images of veneer samples for the experiment were drawn with Adobe Photoshop and Adobe Illustrator design software. The control variable method was used to control the veneer's grain structure, texture and its number of colors unchanged and only change the color scheme under the dimensions of color blending mode and texture; the experimental samples were subjected to a uniform normative treatment, with the image size and view angle adjusted to a uniform state, and the background was controlled to be the same white background, to avoid the redundant details from interfering with the subject's sight.

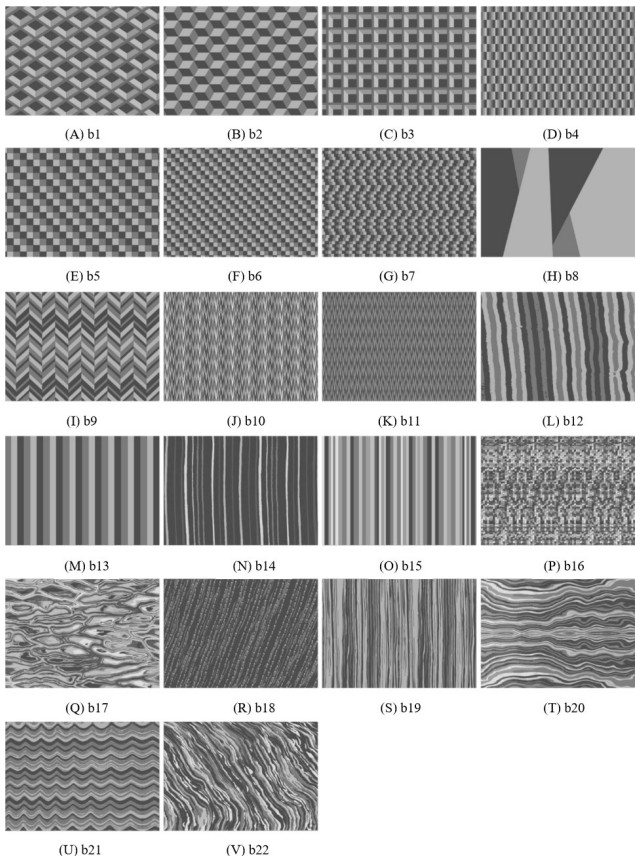

**Figure 9.** Sample images of 22 texture types.

*3.3. Experimental Procedure*

It took about 30 min for the subjects to complete the eye-tracking test, and the experiment was divided into two parts: the preparatory and testing phases. In order to ensure the accuracy of the subjects' gaze data, this study adopts the order of experiment first, then research, to avoid the interference of leaked material during the research on the subjects' eye-tracking test, as shown in Figure 10.

(1)    Preparatory Stage

(1) Familiarize the subjects with the experimental environment. The experimenter introduces the laboratory's equipment, H&S precautions and other related contents to the subjects. (2) Understand the experimental procedure. The experimenter guides the subject to read the experimental procedure and the task description and answers relevant questions. (3) Experiment preparation. Firstly, the experimenter guided the subjects to sit in an appropriate 62∼68 cm range in front of the computer monitor equipped with the eye-tracking device and reminded them to keep their heads as still as possible. Afterward, the experimenter started the eye-tracking device to check whether the subject's line of sight was within the signal acquisition area and whether it could meet the calibration accuracy requirements (deviation of less than 0.5) and adjusted according to the display results. (4) Pre-experimental test. Before the formal experiment, a pre-test should be conducted on the subjects, and the process is consistent with the formal experiment, only replacing the stimulus material. At the end of the pre-test, the results will be recorded to determine whether the subjects could carry out independent experiments, and the results were not adopted and analyzed as the data content of the formal experiment.

(2)    Test Phase

(1) Each subject needs to be calibrated before the formal experiment. The subject's eye position is calibrated first, then five dots will be distributed on the screen, and the subject's

eyeballs need to follow the movement of the dots to calibrate. The poor calibration results will be marked in red, and the five-dot calibration results will be shown in green to indicate that the tracking quality is good and that the experiment can be carried out. (2) Click "Start" to enter the formal experiment. (3) In the formal experiment, a "+" appeared in the center of the computer screen for 1000 ms, and subjects focused on the stimulus. The target image appeared for 1800 ms, then the target image disappeared, and a blank screen was presented for 1800 ms. The above stimuli were repeated for 22 groups, and the experiment was completed. The 22 pages viewed were set in disordered order to eliminate the effect of fixation order on the experimental results. The eye tracker recorded the subjects' eye movement data throughout the experiment. (4) After the subjects completed the experimental task, the experimenter closed the experimental record and saved the data, and the eye movement experiment was completed.

(3)    Filling out The Subjective Questionnaire

Subjects completed the eye movement experiment directly after the questionnaire research of the same experimental material. Each picture in the test will obtain the subject's preference score.

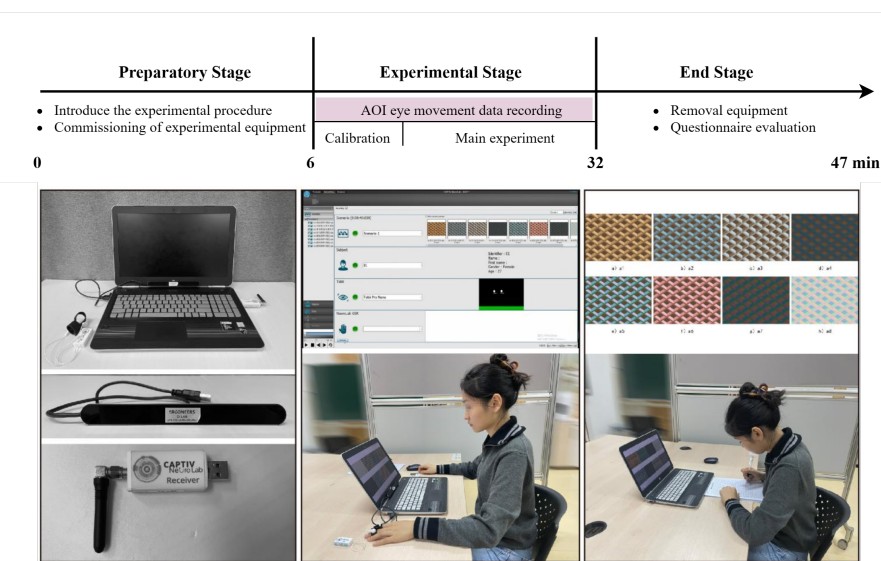

**Figure 10.** Data acquisition equipment and specific process.

## 4. Data Analysis and Discussion

### 4.1. Analysis and Discussion of Objective Experimental Data

A total of 22 experimental samples comprised an average combination of RDV images with different textures and color toning methods. Each experimental sample consisted of eight images, and the data extraction divided the data groups by creating areas of interest (AOIs). The rectangular area corresponding to each veneer image in the sample group image was divided into areas of interest (AOI) and numbered in the eye-tracking software NEUROLAB, as shown in Figure 11. A total of 176 areas of interest were obtained. In the AOI region of interest analysis, three indicators were chosen to output the first gaze time, gaze duration, and gaze number. The experimental data were analyzed in terms of three components: the AOI, the hotspot map, and the time of first gaze.

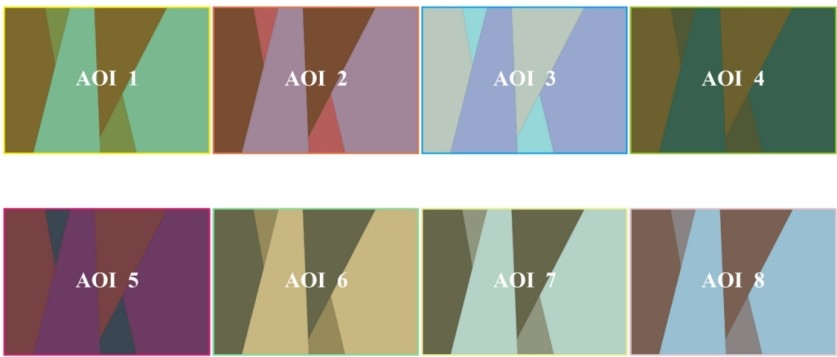

**Figure 11.** An example of AOI region of interest division.

### 4.1.1. AOI Analysis

The extent of the effect of color palette texture type on overall attention was determined by examining the relationship between color palette, texture type and overall level of attention to determine each factor's different levels of significance. Firstly, an analysis of variance was conducted between each factor and the level of attention. If a significant relationship existed, post hoc multiple comparisons could be made to determine further the interaction effect of color palette approach, texture type and attention.

The acquired gaze metrics were examined, and Table 6 illustrates the between-subjects effect tests for the two gaze metrics with the texture and color palette modalities. It can be seen that the significance $p$ of the number of attention and attention duration for different textures are 0.080 > 0.05 and 0.066 > 0.05, respectively, which indicates that there is no significant difference between the attention indicators and texture type, i.e., there is no significant difference in visual stimulation of the subjects by texture type; for significance $p$ = 0.000, the F-value of the number of attention and attention duration for different color toned modality F-value of 71.67 and 60.188, respectively, indicating that the differences in the number of gaze and gaze duration between the color blending modalities are significant, i.e., the color blending modalities have significantly different visual stimuli for the subjects.

Under the number of gazes, the F-value of texture * color blending modality in Table 6 is 2.729. The F-value under gaze duration is even smaller at 1.855, with a significance $p$ = 0.000, which shows that there is a significant interaction between texture and color blending modality, and the combined significance analysis illustrates that the visual stimulation of the subjects on texture will be affected with the increase of the color blending modality.

**Table 6.** Texture types and quantity proportion of artistic engineered veneer.

| Source | Dependent Variable | df | MS | F | *p* |
|---|---|---|---|---|---|
| Texture | Gaze counts | 21 | 84.394 | 1.63 | 0.080 * |
| Color palette | Gaze count | 7 | 3 711.68 | 71.67 | 0.000 *** |
| Texture × bending method | Number of times of gaze | 147 | 141.319 | 2.729 | 0.000 *** |
| Texture | Duration of gaze | 21 | 117.918 | 1.628 | 0.066 * |
| Color palette | Duration of gaze | 7 | 4359.254 | 60.188 | 0.000 *** |
| Texture × color palette | Duration of gaze | 147 | 134.316 | 1.855 | 0.000 *** |

***, * represent 1% and 10% significance levels.

Using NEUROLAB analysis software, the number of times subjects looked at different color palettes and texture types while viewing pictures of restructured decorative thin wood, and the percentage of total duration of gaze were analyzed and summarized, followed by an ANOVA of the two data using SPSS software, and the results are shown in Table 7. Further post hoc multiple comparisons were conducted to examine the relationship between color modulation and gaze metrics. For the variables of the number of gazes and gaze duration, the mean sizes exhibited a high degree of consistency in the following descending order:

a3 > a8 > a1 > a2 > a5 > a6 > a4 > a7 (asymmetric diagonal modulation > high luminance circumferential modulation > vertical modulation > symmetric diagonal modulation > cool-toned helical modulation > warm-tone spiral toning > radial toning > low-luminosity circumferential toning). Specifically, regarding the number of gaze times, significant differences were observed at all levels, except for no significant differences between a1 and a2, a5 (vertical and symmetrical oblique tonality and cool-tone spiral tonality), and between a2 and a5 and a8 (symmetrical oblique tonality and cool-tone spiral tonality and high-brightness circumferential tonality). Concerning gaze time, significant differences were present at all levels except between a1 and a2, a5 (vertical toning vs symmetrical oblique toning and cool-tone spiral toning) and a3 and a8 (asymmetrical oblique toning vs. high luminance circumferential toning), and a4 and a7 (radial toning vs. low luminance circumferential toning), which were not significantly different from one another.

**Table 7.** Multiple posterior comparisons between color harmony mode and eye movement evaluation index.

| Gaze Counts | | | | Gaze Times | | | |
|---|---|---|---|---|---|---|---|
| Palette I | J | Mean Difference I-J | Significance $p$ | Palette I | J | Mean Difference I-J | Significance $p$ |
| a1 | a2 | 0.18 | 0.66 | a1 | a2 | 0.98 | 0.041 ** |
| | a3 | −1.764 | 0.000 *** | | a3 | −1.246 | 0.021 ** |
| | a4 | 4.054 | 0.000 *** | | a4 | 4.77 | 0.000 *** |
| | a5 | 0.816 | 0.049 ** | | a5 | 1.647 | 0.001 *** |
| | a6 | 1.639 | 0.000 *** | | a6 | 2.433 | 0.000 *** |
| | a7 | 4.535 | 0.000 *** | | a7 | 5.046 | 0.000 *** |
| | a8 | −0.197 | 0.64 | | a8 | −0.608 | 0.267 |
| a2 | a3 | −1.944 | 0.000 *** | a2 | a3 | −2.227 | 0.000 *** |
| | a4 | 3.874 | 0.000 *** | | a4 | 3.79 | 0.000 *** |
| | a5 | 0.636 | 0.098 * | | a5 | 0.667 | 0.105 |
| | a6 | 1.459 | 0.000 *** | | a6 | 1.453 | 0.000 *** |
| | a7 | 4.355 | 0.000 *** | | a7 | 4.066 | 0.000 *** |
| | a8 | −0.377 | 0.335 | | a8 | −1.588 | 0.001 *** |
| a3 | a4 | 5.818 | 0.000 *** | a3 | a4 | 6.016 | 0.000 *** |
| | a5 | 2.58 | 0.000 *** | | a5 | 2.894 | 0.000 *** |
| | a6 | 3.403 | 0.000 *** | | a6 | 3.68 | 0.000 *** |
| | a7 | 6.299 | 0.000 *** | | a7 | 6.293 | 0.000 *** |
| | a8 | 1.568 | 0.001 *** | | a8 | 0.639 | 0.243 |
| a4 | a5 | −3.238 | 0.000 *** | a4 | a5 | −3.123 | 0.000 *** |
| | a6 | −2.416 | 0.000 *** | | a6 | -2.337 | 0.000 *** |
| | a7 | 0.481 | 0.020 ** | | a7 | 0.276 | 0.26 |
| | a8 | −4.251 | 0.000 *** | | a8 | −5.378 | 0.000 *** |
| a5 | a6 | 0.822 | 0.023 ** | a5 | a6 | 0.786 | 0.049 ** |
| | a7 | 3.719 | 0.000 *** | | a7 | 3.399 | 0.000 *** |
| | a8 | −1.013 | 0.011 ** | | a8 | −2.255 | 0.000 *** |
| a6 | a7 | 2.896 | 0.000 *** | a6 | a7 | 2.613 | 0.000 *** |
| | a8 | −1.835 | 0.000 *** | | a8 | −3.041 | 0.000 *** |
| a7 | a8 | −4.732 | 0.000 *** | a7 | a8 | −5.654 | 0.000 *** |

***, **, * represent 1%, 5% and 10% significance levels.

### 4.1.2. Hotspot Chart Analysis

Each stimulus image in the experimental material is a specific region of interest, and the hotspot map represents the spatial and temporal distribution characteristics of the eye movement data and is able to indicate the degree of subjects' attention to each region of interest by the difference of colors (from red to yellow and from yellow to green). The red-green gradient indicates the degree of attention, with red representing the highest

number of views, indicating that the subjects paid a high degree of attention to the region; yellow representing a relatively high number of views, indicating that the subjects paid a moderate degree of attention to the region; green representing the region that the subjects paid a low number of views, indicating that the subjects paid an average degree of attention to the region and blank areas indicating regions that did not attract the attention of the subjects at all.

Figure 12 is a visual image that superimposes the hotspot maps of all subjects and is not intended to depict the visual behavior of individual subjects. As can be seen from the hotspot maps, when the subjects were performing observation of the veneer images, the gaze region of the asymmetric oblique tonal, high luminance circumferential tonal showed a large area of red and yellow gradient colors, indicating that this region could significantly attract the subjects' gaze interest. The asymmetric oblique tone and the hue primary-secondary relationship are prominent, and the color and luminance changes are regular. The high brightness of the circumferential palette is consistent with its brightness and color. This is followed by vertical and symmetrical oblique toning, with a small yellow-green gradient in the viewing area. Both of these tonalities are characterized by equal-order changes in luminance and a clear separation of primary and secondary.

In addition, it can be seen from the hotspot map that most of the gaze region of the radial tonus and low-luminance circumferential tonus showed a slightly lighter green color, indicating that the subjects paid average attention to this region. Taken together, it seems that the circumferential tonality, with high luminance, is more likely to attract subjects' eye attention.

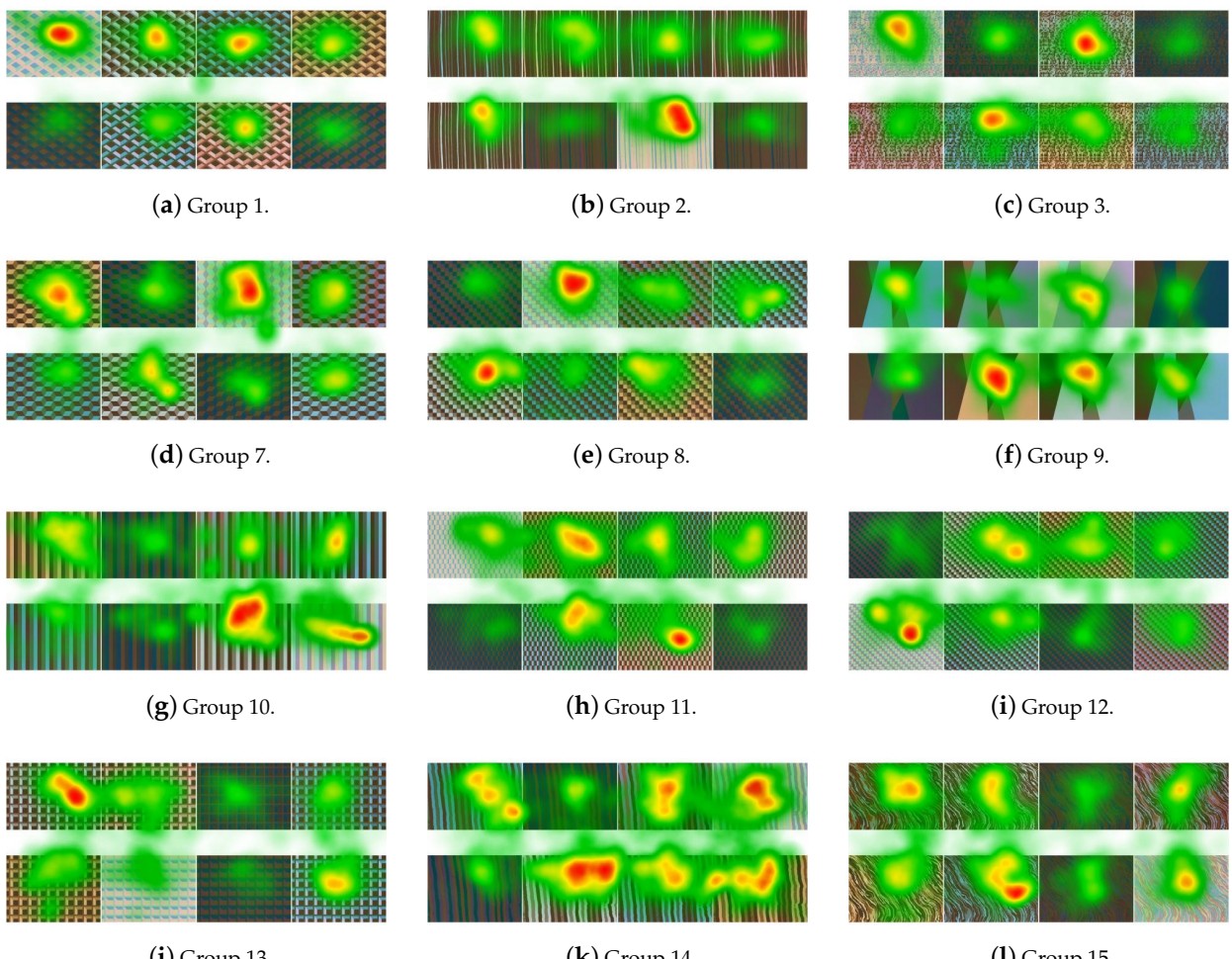

(**a**) Group 1.     (**b**) Group 2.     (**c**) Group 3.

(**d**) Group 7.     (**e**) Group 8.     (**f**) Group 9.

(**g**) Group 10.     (**h**) Group 11.     (**i**) Group 12.

(**j**) Group 13.     (**k**) Group 14.     (**l**) Group 15.

**Figure 12.** *Cont.*

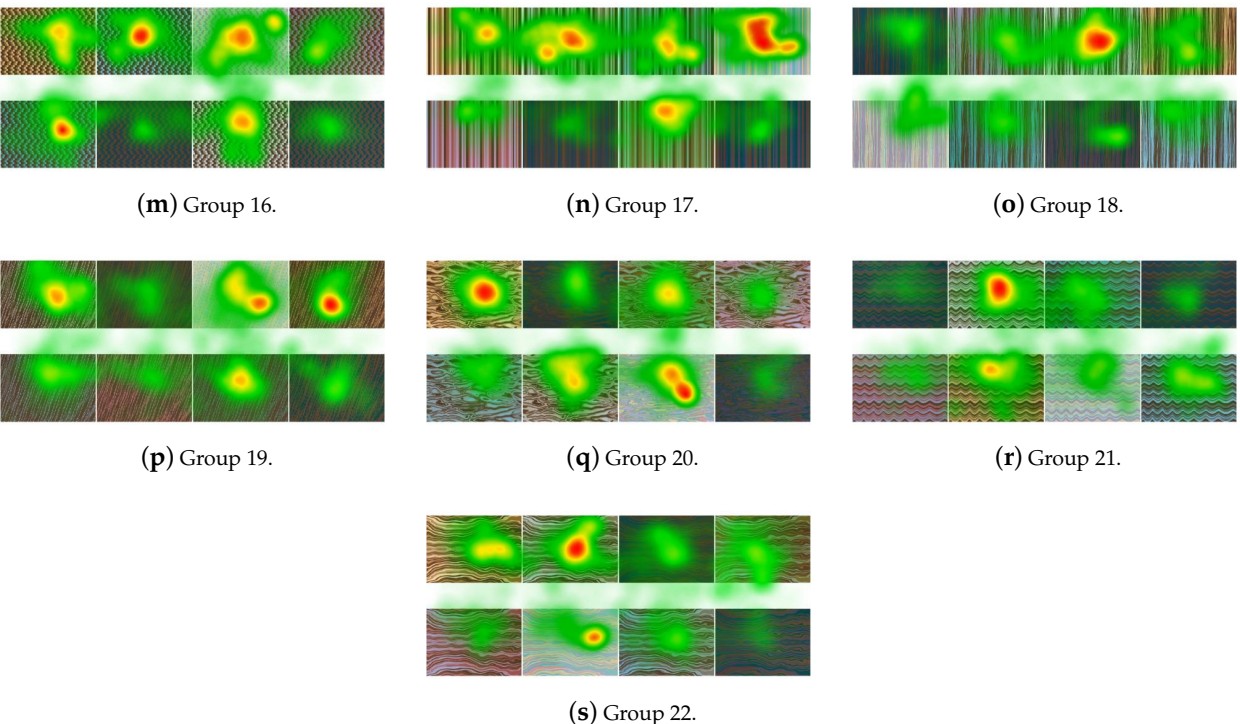

(**m**) Group 16.  (**n**) Group 17.  (**o**) Group 18.

(**p**) Group 19.  (**q**) Group 20.  (**r**) Group 21.

(**s**) Group 22.

**Figure 12.** Grouping thermodynamic diagram.

Although the hotspot map can clearly convey to the researcher the area that subjects pay attention to in the line of sight, and visually reflect the subjects' attention tendency. However, the graphical presentation form cannot quantitatively convey more precise eye movement data, so when comparing the degree of attention of two more similarly colored regions, further analysis should be conducted in conjunction with other quantitative eye movement data.

4.1.3. Analysis of Indicators of First Attention

The first gaze duration refers to a subject's time in the current area of interest when they initially fixate on it. A longer gaze duration indicates that the area of interest can more effectively capture the subjects' attention upon their first glance. It also suggests that subjects require more time to process the information due to the high complexity of the content in the area of interest. Table 8 presents the mean values of first gaze times for all subjects across the 176 areas of interest. Identifying the maximum value in each group reveals the sample that attained the longest first gaze time in that particular group of experimental samples. Notably, the longest first gaze time was observed in the images with the same texture but different color palettes.

As indicated in Table 8, the maximum values within the same experimental group are generally concentrated in the region of interest associated with radial toning and low-luminance circular toning. The radial and low-luminance circumferential toning groups share similar color luminance but exhibit a stark contrast in hue, creating an image that strongly impacts human vision. Exceptions are observed in Group 9 and Group 10, where anomalies are distributed in symmetrical oblique and high brightness circumferential tonality, respectively. The first gaze time is the longest when the texture type remains consistent, and the color tonality is radial or low-lightness circumferential tonality.

**Table 8.** Texture types and quantity proportion of artistic engineered veneer.

| Subject | a1 | a2 | a3 | a4 | a5 | a6 | a7 | a8 |
|---------|-----|-----|-----|-----|-----|-----|-----|-----|
| b1 | 5.8 | 7.8 | 1.8 | 1.1 | 6.3 | 1.5 | 4.5 | 3.5 |
| b2 | 1.4 | 8.1 | 3.2 | 7.5 | 6.3 | 4.6 | 2.3 | 4.6 |
| b3 | 6.2 | 6.4 | 6.3 | 2.4 | 2.1 | 2.7 | 5.6 | 5.4 |
| b4 | 2.7 | 3.4 | 5.8 | 6.7 | 8.7 | 3.8 | 2.4 | 8.3 |
| b5 | 5.3 | 7.7 | 2.5 | 1.7 | 5.9 | 2.5 | 7.2 | 5.7 |
| b6 | 4.8 | 6.5 | 1.7 | 5.3 | 3.8 | 4.2 | 6 | 2.8 |
| b7 | 2.7 | 9.2 | 5.3 | 6.3 | 1.7 | 3 | 6.4 | 4.2 |
| b8 | 5.5 | 9.2 | 4.9 | 7.5 | 6.2 | 2 | 7.2 | 2.4 |
| b9 | 4.3 | 5.3 | 6.1 | 4.5 | 5.5 | 2.9 | 3.7 | 1.7 |
| b10 | 3.5 | 5.1 | 4.8 | 4.3 | 2 | 7.1 | 5.4 | 2.8 |
| b11 | 2.5 | 9.7 | 3.8 | 4.6 | 8.3 | 4.5 | 6.6 | 5.6 |
| b12 | 3 | 9.7 | 6.3 | 3.3 | 5.1 | 5.9 | 4.5 | 7.3 |
| b13 | 6.4 | 5.7 | 4.9 | 3.1 | 9.5 | 6.7 | 5.9 | 2.6 |
| b14 | 6.9 | 3.8 | 4.9 | 5.6 | 9.7 | 5.4 | 3.8 | 3.7 |
| b15 | 6.3 | 6.1 | 5.9 | 4.5 | 6.5 | 6.1 | 4.9 | 6.4 |
| b16 | 2.5 | 8.7 | 2 | 6.4 | 8.7 | 2.9 | 7.4 | 4.1 |
| b17 | 3.8 | 7.1 | 3.5 | 2.1 | 8 | 4.3 | 4.7 | 6 |
| b18 | 5.1 | 5.8 | 7.1 | 2.4 | 9.3 | 7 | 6.5 | 3.9 |
| b19 | 5.4 | 2.9 | 5 | 3 | 7.1 | 2.7 | 5.9 | 5.5 |
| b20 | 3.4 | 5.5 | 5.6 | 5.4 | 7.9 | 3.8 | 2.4 | 4.9 |
| b21 | 4.2 | 7.4 | 6.4 | 2.2 | 6.3 | 5.1 | 3.1 | 7.3 |
| b22 | 3.5 | 4.2 | 6.4 | 1.7 | 8.9 | 6.2 | 5.4 | 7.7 |

Combining the previous analyses of the percentage of total gaze duration, the number of gaze instances, and the hotspot map data, it can be inferred that the reason why the subjects paid the least visual attention to radial toning and low-luminance circumferential toning, despite ranking high in the first gaze duration, is likely because when subjects first encountered images composed of these two types of color toning modalities, the lower luminance of the colors and the blurring of the texture compelled the subjects to spend a longer time comprehending the textures and colors of the pictures. This increased the difficulty in extracting information from this area of interest, resulting in a more extended first gaze duration.

*4.2. Subjective Preferences Survey Data Analysis and Discussion*

The influence of the color palette and texture type on overall liking was assessed by analyzing the relationship between these factors and preference levels. An analysis of variance was initially performed for each factor with preference. If a significant relationship was identified, post hoc multiple comparisons were carried out to understand further the interaction effect of color palette, texture type, and preference.

Subjects' subjective preference metrics were tested, and Table 8 indicates the between-subjects effect test of subjective preference with texture and color palette approach. It can be seen that the significance $p = 0.098 > 0.05$ between texture and subjective preference indicates that there is no significant difference between subjective preference and texture type, i.e., there is no significant difference in subjective preference evaluation of subjects by texture type; the significance $p = 0.000$ between color palette and subjective preference indicates that there is a significant difference in subjective preference among color palettes, i.e., there is no significant difference in subjective preference evaluation of subjects by color palette. They had significantly different subjective preference ratings for the subjects.

For subjective preference, the F-value of texture * color palette in Table 9 is 1.81, with a significance of $p = 0.000$, which shows that there is a significant interaction between texture and color palette, which indicates that the subjective preference of the subjects for the veneer image and texture type will change with the change of color palette. Further post-test multiple comparisons of the relationship between color blending styles and subjective preferences, as shown in Table 8, showed that for subjective preferences, the

means were ranked in descending order as follows: a3 > a1 > a8 > a2 > a5 > a6 > a4 > a7 (Asymmetric oblique blending > Vertical blending > High-luminance peripheral blending > Symmetric oblique blending > Cool-toned spiral blending > Warm-toned spiral blending > Radial blending > Low-luminance peripheral blending > Radial blending). (toning > low brightness circular toning). There are significant differences between the levels of a1 and a2, a8 (vertical tonality and symmetrical oblique tonality and high brightness circumferential tonality), and a2 and a8 (symmetrical oblique tonality and high brightness circumferential tonality), except for no significant differences between the levels of a1 and a2, a8 (vertical tonality and symmetrical oblique tonality and high brightness circumferential tonality).

**Table 9.** Texture types and quantity proportion of artistic engineered veneer.

| Source | Dependent Variable | df | MS | F | *p* |
|---|---|---|---|---|---|
| Texture | Gaze counts | 21 | 2.328 | 1.416 | 0.098 * |
| color palette | Gaze count | 7 | 433.053 | 378.615 | 0.000 *** |
| Texture * bending method | Number of times of gaze | 147 | 2.07 | 1.81 | 0.000 *** |

***, * represent 1% and 10% significance levels.

### 4.3. Analysis of the Correlation between Objective Gaze and Subjective Preference

In order to verify the relationship between objective gaze indicators and subjective preference, the Pearson correlation coefficient method was used to test the three sets of data. Table 10 shows the results of the correlation analysis between gaze duration, number of gazes and subjective preference. It can be seen that: the significance relationship *p* of the 3 groups of data is 0.000, all of them are correlated and significant. Among them, the correlation coefficient between gaze duration and preference is 0.297, which is a weak correlation; the correlation coefficient between the number of times of gaze and preference is 0.299, which is a weak correlation and the correlation coefficient between gaze duration and the number of times of gaze is 0.810, which is a very strong correlation.

**Table 10.** Correlation analysis of subjective and objective data.

| Variables | Relevance | Preference | Graze Counts | Graze Times |
|---|---|---|---|---|
| Preference | Pearson | 1 (0.000 ***) | | |
| Graze Counts | Pearson | 0.299 (0.000 ***) | 1 (0.000 ***) | |
| Graze Times | Pearson | 0.297 (0.000 ***) | 0.81 (0.000 ***) | 1 (0.000 ***) |

*** represents 1% significance levels.

## 5. Conclusions

This study adopts a combination of eye-tracking tests and subjective preference research. Firstly, three objective indicators of subjects' gaze duration, the number of gaze times and time of first gaze were used to analyze and compare subjects' visual attention to images of RDV with different color palettes and texture types in conjunction with hotspot diagrams. Secondly, the subjective questionnaire data were quantitatively evaluated and compared. Once again, combining subjective and objective data, the relationship between objective eye movement indexes and overall evaluation was verified, and the reliability and accuracy of the analysis conclusions were clarified. The results of this experiment provide scientific and rational experimental reference bases for assisting in the design of the color-mixing methods and texture types of RDV. This experiment obtained the following conclusions based on the data analysis:

(1) There is a high degree of consistency between the subjects' gaze duration, gaze counts and hotspot maps. There is a significant difference between the gaze indicators and the color mixing modes, amongst which asymmetric oblique mixing, high luminance circumferential mixing, and vertical mixing are ranked at the top of the gaze indicators and all receive a high degree of attention. In contrast, radial mixing and low luminance circumferential mixing are ranked at the bottom of the gaze indicators and receive less

attention. In the first gaze time, the maximum value is generally distributed in the interest area of radial tonality and low luminance circumferential tonality. Combining the results of the first three gaze indicators with the later subjective questionnaire study, it can be found that this result is caused by the fact that the colors in the radial tonality and low luminance circumferential tonality color groups are all low luminance and have a single level of color structure, which results in the subjects not being able to differentiate between the texture and the colors in the experimental images. The difficulty in extracting relevant color and texture information increased, and the first gaze time was longer.

(2) Subjective preference for the experimental samples differed significantly from the color blending method, while the relationship with texture type was insignificant. The experimental images with asymmetric oblique toning, high luminance circumferential toning, and vertical toning had the highest subjective ratings and degree of liking. In contrast, the experimental images with low luminance circumferential and radial toning had the lowest subjective ratings and lower degrees of liking. This result has some commonality with the results of the eye-tracking test, which laterally reflects that the gaze metrics can quantify the visual attractiveness of the RDV images to a certain extent.

(3) The correlation analyses among the three data groups, namely, gaze duration, gaze counts and preference, all showed positive correlations. Among them, the correlation between the two attention indicators of gaze duration and number of times of gaze showed significant differences. The analysis of the common law of combining images with a high degree of attention revealed that the stronger the stimuli, such as high brightness, strong contrast, and color hierarchy, the higher the level of attention was. However, the correlation between the two attention indicators and the subjective preference scores is average, indicating that the objective attention indicators do not accurately reflect the subjects' preferences. It would be more effective to analyze the subjects' attention and preference by combining the eye-tracking measurement technique and the subjective evaluation method when evaluating the visual perception of the RDV.

This study, delving into the color and texture characteristics of RDV from the perspective of visual perception, offers valuable insights for the development and design of new products in this field. However, there are some limitations and challenges that should be acknowledged:

(1) Limited the Population and Experimental Material: The main population in the experiment was set between the ages of 20~26 years old, which could be more relaxed and subdivided into age groups in subsequent studies. It is important to pay attention to the preference bias of different age groups. In the form of experimental materials, pictures were used to differentiate between colors and textures. In the subsequent research, can use the physical form, increase the texture and other conditions to enrich the choice of experimental materials, closer to the actual life.

(2) Limited Experimentation Variables: The visual perception experiment utilized eye-tracking devices for data acquisition. To enhance the scientific rigor, future studies might consider incorporating additional parameters such as EEG and heart rate. Moreover, the experiment focused on two variables—color mixing mode and texture type. Exploring other potential influencing variables like color area, color position relationship, and texture size could provide a more nuanced understanding of users' visual perception.

(3) Limited Theoretical Research Depth: The theoretical research section has a relatively small number of studies related to visual perception, and there is room for improvement in the depth of research. Strengthening the foundational theoretical knowledge and expanding the scope of related literature studies could enhance the overall theoretical framework.

(4) Multidisciplinary Complexity: The design of recombinant decorative thin wood involves a multidisciplinary approach, integrating art, timber science, and furniture manufacturing technology. While this study focuses on color and texture, other crucial factors such as gloss characteristics, scale, and application range have not been thoroughly explored. Future research could delve into these aspects to ensure a comprehensive understanding.

**Author Contributions:** T.H. contributed to Visualization and Writing-original draft. C.Z. contributed to Data curation, Funding acquisition and Writing—review and editing. X.W. contributed to Visualization, Data curation and Methodology. J.K. contributed to Writing—review and editing. All authors have read and agreed to the published version of the manuscript.

**Funding:** Part of this work was supported by the 2020 Jiangsu Postgraduate "International Smart Health Furniture Design and Engineering" project, and 2022 Jiangsu Province Ecological Health Home Furnishing Industry-University-Research International Cooperation Joint Support for laboratory projects. Part of this work was also sponsored by Qing Lan Project.

**Institutional Review Board Statement:** Not applicable.

**Informed Consent Statement:** Informed consent was obtained from all subjects involved in the study. Written informed consent has been obtained from the patients to publish this paper. The study was conducted in accordance with the Declaration of Helsinki, and the protocol was approved by the Ethics Committee of Nanjing Forestry University.

**Data Availability Statement:** Data are contained within the article.

**Acknowledgments:** This study was supported by the Scientific Research Support project provided by Kingfar International Inc. Thanks for the research technical and ErgoLAB Man-Machine-Environment Testing Cloud Platform (ErgoLAB v3.0) related scientific research equipment support of Kingfar project team.

**Conflicts of Interest:** The authors declare no conflict of interest.

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
