# Peer review of "A Study of Visual Perception Based on Colour and Texture of Reconstituted Decorative Veneer"

_coatings, doi:10.3390/coatings14010057_

Round 1
Reviewer 1 Report
Comments and Suggestions for Authors
The authors concentrate on the user’s visual perception of Reconstituted Decorative Veneer, especially into extracting and analyzing the color and texture features of Reconstituted Decorative Veneer and the experimental study of evaluating the combination of color and texture. In the visual perception research, it was concluded that the design of the join of color and texture has a notable result on the objective eye movement data.
The paper is interesting and fits the profile of the Coatings journal. The layout of the paper is well thought out and the form follows the rules of art. I have no fundamental objections to the content of the paper, but I found some errors in this manuscript, and it must be improved.
Noticed errors:
1. Let the authors decide whether they use UK English or US English. For example: Table 1 and Table 2 captions contradict the words in the previous paragraph (color / colour).
2. Chapter 1 lacked a clear formulation of the research gap on state of art basis.
3. No conclusions for further research. Adding such conclusions will certainly increase the value of the paper.
Small errors
(I hope typographic only) do not reduce the value of the paper, but they must be corrected.
1. Line 70: One veneer is enough …
2. Line 290. Is: … of The numer …; should be: … of the number …
3. Line 617. Is: … images, The … ; should be: … images. The …
Author Response
Thank you very much for your affirmation and valuable comments . In response to your comments, we have made corresponding changes in the article, marked in yellow, hoping to help you better position.
- First of all, thank you very much for mentioning the British English writing style and expression, which is very helpful to us. It also standardises our writing, which is much appreciated. Therefore, the expression of the whole text has been reworked, and we hope it can help you read it better.
- The chapter 1 is a compendium of our current perceptual approach to RDV evaluation, which has been revised.
- We have added our limitations and perspectives on this study in the final section.
Finally, we have corrected all the detailed errors you found in the article. Thank you very much for your careful revision and pointing out.
Reviewer 2 Report
Comments and Suggestions for Authors
The present work contains an elaborated study regarding the analysis of visual perception based on color and texture of reconstituted decorative veneer. It was investigated the relationship between color and visual perception as an evaluation criterion in the design of home finishes. In additionally, it was improving RDV’s decorative properties and visual aesthetics.
Also, that are well described the methodological elaboration for the visual analysis of RDV on the basis of collecting RDV samples, classifying and screening them. It was studied the colour distribution laws and characteristics of different types of veneer and their colour compositions.
However, my comments and recommendations are the following:
1. It was difficult for me to understand well what the authors described. The paragraphs are very long and the essence is lost. It must to be improved.
2. The abstract should be explained in more comprehensively. In my opinion is to longue.
3. Line 76 - Figure 1 is not proper described. In general, the technological processes are defined by the flowchart, not by a picture.
4. Line 55 – I think that the references citations are not in accordance with journal template – It must: [5-7].
5. Line 197 – “The project …”. I suggest “the present paper …”
6. Line 197 – 201. The sentence is not clear.
7. Line 204: “we study their colour distribution low …”. -- > “it was study colour distribution” .
8. Line 206: - “…matching laws…” -- > “adequevate repartition…”
9. Parameters defined in all table, it must be detailed.
10. Line 315:- “In the distribution of lightness …” The “distribution term is not correct used. The distribution means STATISTICAL REPARTITION. I suggest to be replaced with proper terms.
11. Line 433:-“… average distribution ..” – The same – it is not proper used!!!
12. Figure 11 and Figure 12 – are not relevant for proposed study.
13. In section 4.1: - In statistically analysis, the confidence interval (CI) is not defined. I suppose that p = 5%
14. Section 4.2: “Hotspot Chart Analysis” - I have not heard about this statistical analysis! For the veracity of the analysis, some bibliographical references can be mentioned.
15. Additionally, including some conclusions for further researches in the field, it would certainly increase the value of the paper.
Comments on the Quality of English Language
1. It was difficult for me to understand well what the authors described. The paragraphs are very long and the essence is lost. It must to be improved.
Author Response
|
Review |
Response |
|
The present work contains an elaborated study regarding the analysis of visual perception based on color and texture of reconstituted decorative veneer. It was investigated the relationship between color and visual perception as an evaluation criterion in the design of home finishes. In additionally, it was improving RDV’s decorative properties and visual aesthetics. Also, that are well described the methodological elaboration for the visual analysis of RDV on the basis of collecting RDV samples, classifying and screening them. It was studied the colour distribution laws and characteristics of different types of veneer and their colour compositions. It was difficult for me to understand well what the authors described. The paragraphs are very long and the essence is lost. It must to be improved. |
Thank you very much for your acknowledgement and valuable comments on this study. The article does have some expressions that are rather obscure and difficult to understand in the writing. Therefore extensive language touch-ups and revisions were made. We hope that it will not cause you any confusion when you read it again. In addition, the changes you have made are very valuable. In the original text, the locations of the corresponding changes have been marked in yellow, and we hope that this will help you to locate them better. |
|
2. The abstract should be explained in more comprehensively. In my opinion is to longue. |
The summary section has been modified to become shorter and more concise. |
| 3. Line 76 - Figure 1 is not proper described. In general, the technological processes are defined by the flowchart, not by a picture. |
Thank you very much for your correction, it is very helpful. After consulting with experts and scholars in the field, and in conjunction with our research. This part is not very relevant, so we have made deletions. |
| 4. Line 55 – I think that the references citations are not in accordance with journal template – It must: [5-7]. |
Thank you for your questions about our formatting issues, and we have made changes to the relevant issues. |
| 5. Line 197 – “The project …”. I suggest “the present paper …” |
Thank you for raising the issue of our expression, and we have revised the relevant issues. |
| 6. Line 197 – 201. The sentence is not clear. |
Thank you for raising the issue of our expression, and we have revised the relevant issues. |
| 7. Line 204: “we study their colour distribution low …”. -- > “it was study colour distribution” . |
Thank you for raising the issue of our expression, and we have revised the relevant issues. |
| 8. Line 206: - “…matching laws…” -- > “adequevate repartition…” |
Thank you for raising the issue of our expression, and we have revised the relevant issues. |
| 9. Parameters defined in all table, it must be detailed. |
We have annotated the symbols that appear in the table. |
| 10. Line 315:- “In the distribution of lightness …” The “distribution term is not correct used. The distribution means STATISTICAL REPARTITION. I suggest to be replaced with proper terms. |
Thank you very much for the heads up, we also think distribution was not used properly here, hence the deletion. |
| 11. Line 433:-“… average distribution ..” – The same – it is not proper used!!! |
Thank you very much for pointing out that in the expression and context here we replace distribution with combination. |
| 12. Figure 11 and Figure 12 – are not relevant for proposed study. |
Because of the reordering, the previous figure 11 serial number becomes figure 10 and figure 12 serial number becomes figure 11. In fact, Figure 10 is a graphical representation of the entire experimental steps, which can better help readers understand the experimental steps. Figure 10 is the AOI graph after performing the eye movement experiment, which is related to the experimental data. |
| 13. In section 4.1: - In statistically analysis, the confidence interval (CI) is not defined. I suppose that p = 5% |
Thank you very much for pointing this out, and we have added and amended it in the relevant table. |
| 14. Section 4.2: “Hotspot Chart Analysis” - I have not heard about this statistical analysis! For the veracity of the analysis, some bibliographical references can be mentioned. |
Thank you very much for your comments, so we have revised the structure of Section 4 for better reading and understanding. Currently, Chapter 4 is divided into three subheadings, namely "Analysis and Discussion of Objective Experimental Data", "Subjective Preferences Survey Data Analysis and Discussion"," Analysis of The Correlation between Objective Gaze and Subjective Preference". The Hotspot Chart Analysis you mentioned belongs to the "Analysis and Discussion of Objective Experimental Data" section, where our data is the Hotspot Chart, refer to Figure 12. |
| 15. Additionally, including some conclusions for further researches in the field, it would certainly increase the value of the paper. |
Thank you very much for your comments, and we have included additional research limitations and shortcomings in the summary section. |
Reviewer 3 Report
Comments and Suggestions for Authors
This study deals with the investigation of the visual perception of wood surface images displayed on computer screens. This study is very extensive and particularly interesting for the presentation of surfaces on the websites. However, based on the results the conclusion was limited and only the results of the eye-tracking test were used for the comparison. Therefore, the paper is inconsistence and the read thread is missing. In the introduction section, the topic of the eye-tracking test/method was not really picked out as a central theme.
The aim of this study was to research the theory and implementation of visual perception in the development of RDV (reconstituted decorative veneer) (line 197). The production process of RDV was shown in Figure 1 (which was not correctly presented). At the end this paper does not contribute to the fulfilment of this mentioned objective. The reader of this manuscript will little known for the development of RDV. The analysis was made with digital images of the RDV and the conditions (e.g. light, temperature, ...) of the recording was not known for all images. Therefore, what is the reference value to compare your results with other studies?
Furthermore, please check the arrangement of the Figures and the assignment of the labelling and sentences.
Author Response
Thank you very much for your acknowledgement of this study and your valuable comments. This study focuses on the colour and texture of RDVs using objective eye tracking data combined with subjective preference ratings. The main study is on the colour and texture of RDV, and the images chosen are electronic images. What you have submitted is also one of the limitations of the article at this stage, as it is not possible to experiment with physical objects due to the conditions. The light, temperature you mentioned is a new inspiration for us and we will further explore these factors conditions. Changes to the article have been highlighted in yellow to help you better orientate yourself.
Round 2
Reviewer 2 Report
Comments and Suggestions for Authors
The paper can be accepted in present form!
Author Response
Thank you very much for your valuable suggestions regarding this article. Your acknowledgement of the content of this research work is also highly appreciated. Your comments will also be very helpful for our subsequent research.
Reviewer 3 Report
Comments and Suggestions for Authors
The authors reworked the manuscript. The red thread can be seen. The study deals with the analysis of colour and structure of artificial wooden surfaces.
Figure 2 shows the 13280 samples of designed artistic veneer. But is it really 13280 samples or only 280 samples? In line 221, you mentioned that “280 reconstituted artistic modelling veneer samples images (as illustrated in Figure2) were compiled” are used. Where and how were the other 13,000 samples collected?
Please indicate the l1, l2, a1, a2, b1, b2 variables in equation 1;
Line 298: What are the HCL values? Did you mean the h, C, and L values from the CIEL*a*b colour space? Why is it important to change the variable?
Add some additional limitations of this study in the conclusion section:
i) restricted population (only persons, how are 20 to 26 years old)
ii) using only pictures from reconstituted decorative veneers
iii) non-calibrated colour displays for the visual presentation of the colour pictures
Author Response
The authors reworked the manuscript. The red thread can be seen. The study deals with the analysis of colour and structure of artificial wooden surfaces.
Response: Thank you very much for your recognition and your valuable comments. The parts of the article that have been modified are all highlighted in the hope that it will be easier for you to locate the relevant content.
- Figure 2 shows the 13280 samples of designed artistic veneer. But is it really 13280 samples or only 280 samples? In line 221, you mentioned that “280 reconstituted artistic modelling veneer samples images (as illustrated in Figure2) were compiled” are used. Where and how were the other 13,000 samples collected?
Response: Thank you very much for your seriousness and care. After reviewing with the original we found that there was a compilation error in the title here, and the actual picture case is just 280 samples. Thank you very much for your careful checking and review.
- Please indicate the l1, l2, a1, a2, b1, b2 variables in equation 1;
Response: Thank you very much for pointing out this about our specification, we have added this in the text. You can find the relevant additions on line 208.
- Line 298: What are the HCL values? Did you mean the h, C, and L values from the CIEL*a*b colour space? Why is it important to change the variable?
Response: Thank you very much for pointing this out, it makes us realise that this part is not very clearly stated. HCL (Hue-Chroma-Luminance) is a colour space, like RGB, and is also known as CIELch (uv) because it was first proposed by the International Commission on Illumination (CIE). Using the HCL colour space at the colour matching stage allows direct control of perceived brightness, making it easier to achieve a high degree of usability in product colour matching. Due to space constraints this article is explained in lines 299 to 303 to help you understand the process.
- Add some additional limitations of this study in the conclusion section:
i) restricted population (only persons, how are 20 to 26 years old)
ii) using only pictures from reconstituted decorative veneers
iii) non-calibrated colour displays for the visual presentation of the colour pictures
Response: Thank you very much for your comments, which we have added to the final limitation of the article. The age limitation and the choice of the form of experimental material is a pity for us, and we will improve our content by considering it more comprehensively in our future work.
